# Deep learning and single-cell phenotyping for rapid antimicrobial susceptibility detection in Escherichia coli

Alexander Zagajewski [1,2,7], Piers Turner[1,2,7], Conor Feehily[3], Hafez El Sayyed[1,2], Monique Andersson[3,4], Lucinda Barrett[4], Sarah Oakley[4], Mathew Stracy[5], Derrick Crook[3,4], Christoffer Nellåker [6✉], Nicole Stoesser [3,4✉] & Achillefs N. Kapanidis [1,2✉]

The rise of antimicrobial resistance (AMR) is one of the greatest public health challenges, already causing up to 1.2 million deaths annually and rising. Current culture-based turnaround times for bacterial identification in clinical samples and antimicrobial susceptibility testing (AST) are typically 18–24 h. We present a novel proof-of-concept methodological advance in susceptibility testing based on the deep-learning of single-cell specific morphological phenotypes directly associated with antimicrobial susceptibility in *Escherichia coli*. Our models can reliably (80% single-cell accuracy) classify untreated and treated susceptible cells for a lab-reference fully susceptible *E. coli* strain, across four antibiotics (ciprofloxacin, gentamicin, rifampicin and co-amoxiclav). For ciprofloxacin, we demonstrate our models reveal significant ($p < 0.001$) differences between bacterial cell populations affected and unaffected by antibiotic treatment, and show that given treatment with a fixed concentration of 10 mg/L over 30 min these phenotypic effects correlate with clinical susceptibility defined by established clinical breakpoints. Deploying our approach on cell populations from six *E. coli* strains obtained from human bloodstream infections with varying degrees of ciprofloxacin resistance and treated with a range of ciprofloxacin concentrations, we show single-cell phenotyping has the potential to provide equivalent information to growth-based AST assays, but in as little as 30 min.

[1] Department of Physics, University of Oxford, Parks Road, Oxford OX1 3PJ, UK. [2] Kavli Institute for Nanoscience Discovery, University of Oxford, South Parks Road, Oxford OX1 3QU, UK. [3] Nuffield Department of Medicine, University of Oxford, John Radcliffe Hospital, Oxford OX3 9DU, UK. [4] Department of Microbiology and Infectious Diseases, Oxford University Hospitals NHS Foundation Trust, Oxford OX3 9DU, UK. [5] Sir William Dunn School of Pathology, University of Oxford, South Parks Road, Oxford OX1 3RE, UK. [6] Nuffield Department of Women's & Reproductive Health, University of Oxford, Big Data Institute, Oxford OX3 7LF, UK. [7] These authors contributed equally: Alexander Zagajewski, Piers Turner. ✉email: christoffer.nellaker@wrh.ox.ac.uk; nicole.stoesser@ndm.ox.ac.uk; achillefs.kapanidis@physics.ox.ac.uk

Antimicrobial resistance (AMR) is a major public health challenge, causing an estimated 1.2 million deaths annually[1], with this number predicted to rise much further if left unchecked. AMR represents the evolutionary effect of antimicrobial selection pressures in the context of the short bacterial cell cycle, leading to adaptation by natural selection through a variety of molecular mechanisms[2]. Several clinical strategies to address the AMR crisis have been considered. One strategy relies on the continuous development of novel antimicrobial agents to outpace bacterial evolution; however, this strategy alone is neither scientifically nor economically viable[3,4]. Another strategy relies on the conservation of the existing antimicrobial arsenal through strict stewardship and regulation, which remains challenging to implement universally[5]. A third option, as part of stewardship, is through diagnostic improvements, including more rapid Antimicrobial Susceptibility Testing (AST) methods, which allow better tailoring of antibiotic treatment regimens given to patients[6]. A combination of these and other approaches, such as vaccination, is likely needed, as no single strategy currently represents a complete solution across all settings.

Existing ASTs provide phenotypic quantification of the Minimum Inhibitory Concentration (MIC) of an antibiotic of choice for isolates cultured from infected patients, and can be complemented by targeted nucleic-acid assays for known resistance determinants. AST may also be preceded by species identification (e.g., using MALDI-ToF). Several drawbacks of current bacterial identification and AST workflows are a requirement for culture-based isolation of clinical pathogens, expert operators and laboratory space, and a typical turnaround time of 18–24 h from sampling to results[7], although faster growth-based AST platforms are being more widely deployed (e.g. Accelerate's Pheno System, bioMerieux's SPECIFIC REVEAL® Rapid AST System). Initial antimicrobial regimens given to sick patients are therefore usually broad-spectrum, which may maximise collateral patient-level effects such as perturbation of gut flora, and contribute to the selection and dissemination of AMR at both the patient- and population-levels.

Multiple novel approaches to improve the speed of AST exist, including biosensors, genomic assays, and hybridisation approaches;[8] however, most of these remain in the development stage[7], and have not yet been translated into practice. Many of these methods assess an *entire* bacterial culture, and the lack of single-cell specificity leaves them insensitive to heterogeneity in cell populations, such as the presence of persister cells, or polymicrobial mixtures. A potential solution to this problem is the use of single-cell specific ASTs, which address the heterogeneity problem while also offering higher throughput by evaluating the effect of antimicrobials *directly* on cells, rather than relying on secondary markers such as growth of an entire culture. Multiple such candidate ASTs have been proposed, enabled by platforms such as flow cytometry[9], Raman spectroscopy[10,11], fluorescent probes in droplet microfluidic devices[12], impedence cytometry[13] and others. Further opportunity in single-cell ASTs comes from their integration with widefield microscopy[14–17], which increases throughput by enabling real-time simultaneous monitoring of large numbers of individual cells. Such cellular imaging produces rich, high-volume, unstructured data that are well suited to machine-learning based analysis, and in particular, by modern deep-learning techniques. These techniques have been used to great effect to produce AST inference models from genomic[18,19] and metabolomic[20–23] data, where their ability to execute their own feature engineering maximises the usage of complex, unstructured data. With similar insights, deep-learning has been applied to widefield microscopy to produce candidate ASTs that provide phenotypic quantification by monitoring single-cell growth[24,25] or motion patterns[26,27] in the presence of antibiotics.

Another microscopy approach relies on directly evaluating the effect of antimicrobials on cellular structures, such as the bacterial nucleoid or the cell membrane. These structures have been characterised experimentally and computationally[28,29], and were used as single-cell phenotypes to profile cytological pathways to understand the mode of action of antibiotics[30], and to test for methicillin resistance in *Staphylococcus aureus*[31]. A wide range of nucleoid and cell membrane cytological phenotypes under different treatment conditions have since been established for a range of Gram-positive and Gram-negative species[32–34]. Such a phenotyping approach has notable advantages over single-cell microscopy assays monitoring growth or motion patterns: results are available on the timescale of a single bacterial life cycle rather than several lifecycles, the method is applicable to difficult-to-culture pathogens, and there is no requirement for continuous tracking of individual live cells over time. However, cellular phenotyping may be affected by phenotypic plasticity, whereby small genotypic and environmental differences can strongly influence the displayed phenotype.

In this work, we introduce a novel, single-cell, microscopy-based approach to characterising bacterial antibiotic susceptibility that in principle could address some shortcomings of existing assays. We combine single-cell phenotypes of the nucleoid and cell membrane with modern Convolutional Neural Networks (CNNs) to develop a method based on deep phenotyping of individual cells that display different physiological responses to antibiotics. CNNs feature a hierarchical pattern of learnable convolution filters, which allows efficient learning in heterogenous imaging data without manual feature engineering – removing the main bottleneck of previous work[31]. Our Deep Antimicrobial Susceptibility Phenotyping (DASP) platform uses widefield micrographs to rapidly classify antibiotic-treated cells as either susceptible or resistant. We have developed specific models for four antibiotics, each representative of an antibiotic family with a different mode of action: the fluoroquinolone ciprofloxacin (which targets DNA synthesis), the aminoglycoside gentamicin (targets protein synthesis), the beta-lactam co-amoxiclav (targets cell-well synthesis) and rifampicin (targets RNA synthesis). We show our models appear robust to phenotypic plasticity by training them on a lab strain of *Escherichia coli* and then deploying them successfully on several *E. coli* clinical isolates with different ciprofloxacin MICs, where we were able to distinguish bacteria displaying changes in cellular phenotype in response to fixed ciprofloxacin treatment concentrations. Furthermore, we demonstrate that for ciprofloxacin, varying the treatment concentration generates a dose-response relationship that might allow precise quantification of the MIC of clinical isolates over timeframes as little as 30 min.

## Results

### Detecting antibiotic susceptibility based on deep learning of single-cell subcellular phenotypes.
We designed a method that takes as an input bacterial cultures grown in rich medium to a consistent optical density and then treated with an antibiotic of choice for a time sufficient to produce distinct, antibiotic-specific, cellular phenotypic changes (Fig. 1A, left) - in which case bacteria were classified as "susceptible" to the given antibiotic. Upon cell fixation and staining (Fig. 1A), cells that remained unaffected by the treatment were classified as having a "resistant" cellular phenotype, which highly resembled the original untreated phenotype (see Supplementary Fig. S1 for example comparing the resistant and untreated phenotypes in resistant clinical isolates).

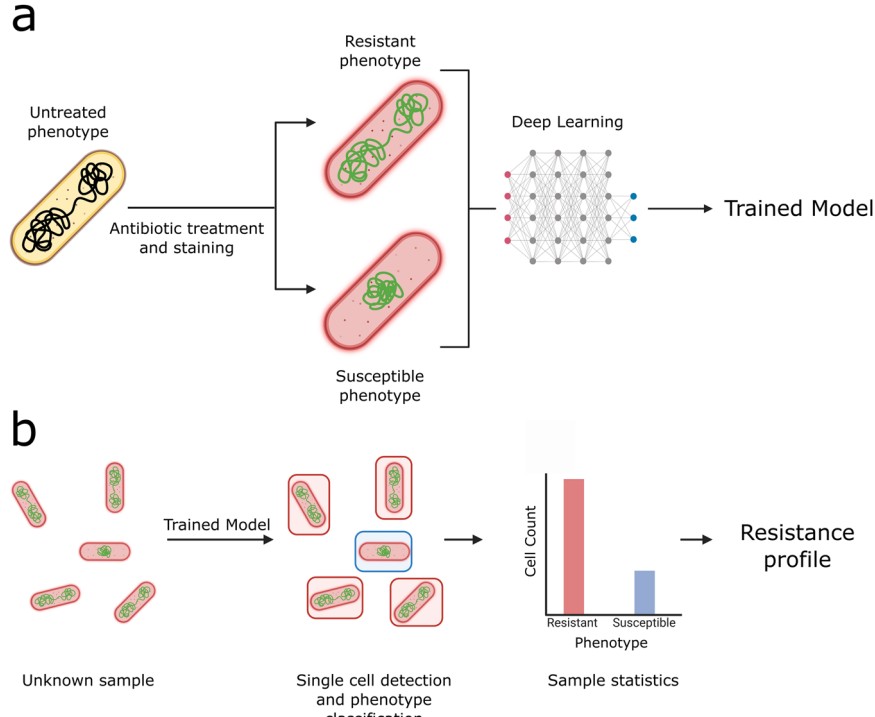

**Fig. 1 Schematic of an approach to antimicrobial susceptibility testing based on bacterial single-cell phenotypes. A** Live *E.coli* cells are treated with an antibiotic, inducing changes in subcellular morphology. Susceptible cells show strong phenotypic changes associated with the effect of antibiotic action, creating a distinct susceptible phenotype. Resistant cells are not affected by the antibiotic – the resistant phenotype is similar to the untreated phenotype. Cells are fixed, and nucleoids and cell membranes are fluorescently stained. The sample is imaged under a widefield fluorescence microscope. A deep learning pipeline is trained to distinguish the susceptible phenotype from the resistant (untreated) phenotype, with single cell resolution. **B** An unknown sample can be processed and fed into the trained model, which classifies the phenotypes on a single cell level, to produce sample-wide classification statistics. These statistics can then be used to obtain information on the resistance of the entire sample.

After collecting a large number of micrographs (each representing an image containing 50–200 cells), individual cells representing either the untreated or susceptible phenotype were used to train segmentation and classification models. At testing, the trained models segment micrographs of treated cells and classify individual cells into one of the two categories with regards to antibiotic susceptibility; examining the distribution of classifications enables reporting on the resistance of the entire population of bacterial cells making up the sample (Fig. 1B).

**Generating antibiotic-resistant and antibiotic-susceptible cellular phenotypes**. To implement the concept above, we characterised the untreated cellular phenotype of lab reference *E. coli* strain MG1655, as well as the susceptible cellular phenotypes of the same strain to four antibiotics reflecting different modes of action: ciprofloxacin (CIP; MG1655 MIC = 0.012 mg/L), gentamicin (GENT; MG1655 MIC = 0.13 mg/L), co-amoxiclav (COAMOX; MG1655 MIC = 3.2 mg/L), or rifampicin (RIF; MG1655 MIC = 8 mg/L), where CIP, GENT and COAMOX are widely used in clinical practice as treatment for infections caused by *E. coli* (see *Methods* and Supplementary Table S1). Rifampicin is not clinically relevant for *E. coli*. The MG1655 *E. coli* strain was incubated with supra-MIC concentrations of 10 mg/L, 40 mg/L, 100 mg/L and 160 mg/L of ciprofloxacin, gentamicin, rifampicin and co-amoxiclav for 30 min, 30 min, 30 min and 60 min respectively, in order to capture discernible changes in cellular morphology. The Amoxicillin and clavulanic acid co-amoxiclav in co-amoxiclav were prepared at a 2:1 ratio (The clavulanic acid concentration was 80 mg/L at the co-amoxiclav treatment concentration). To capture the untreated cellular phenotype (and use it as a proxy for the resistant phenotypes, where we expect no

treatment-induced changes in the nucleoid and membrane) and the susceptible phenotype for each antibiotic, we stained the bacterial nucleoid with the DNA-binding fluorophore DAPI (green signals in Fig. 2A–E), and the membrane with the lipid stain Nile Red (NR; red signals in Fig. 2A–E), revealing the organisation of the nucleoid and overall cell morphology as a function of treatment.

In the untreated phenotype (Fig. 2A), distinct copies of the chromosome were seen in each cell, organised into heterogenous macrodomains by nucleoid-associated proteins[35]. In contrast, incubating MG1655 with 10 mg/L of ciprofloxacin produced a compaction of the chromosome towards the cell centre due to topoisomerase IV inhibition[36] (Fig. 2B). Similarly, incubation with gentamicin, which binds to the 30S ribosomal subunit and interferes with translation elongation, also leads to nucleoid compaction, although the chromosomes in this case do not merge fully into one spot (Fig. 2C). Incubation with rifampicin, which inhibits transcription initiation by RNA polymerase, led to decompaction of the nucleoid (Fig. 2D); individual chromosomes could still be distinguished, but the macrodomains were lost. Finally, exposure to co-amoxiclav produces a subtle susceptible phenotype – whilst some differences in the organisation of the macrodomains can be seen, the effect is more challenging to discriminate visually from the untreated phenotype. Representative full fields of view for all phenotypes are provided in Supplementary Figs. S2 and S3.

To classify the phenotypes, we designed a 2-stage deep-learning pipeline (Fig. 2F). In the first stage, a Mask-Region based Convolutional Neural Network (RCNN) model[37] segments individual cells from whole micrographs using the image generated using NR stain. In the second stage, a separate

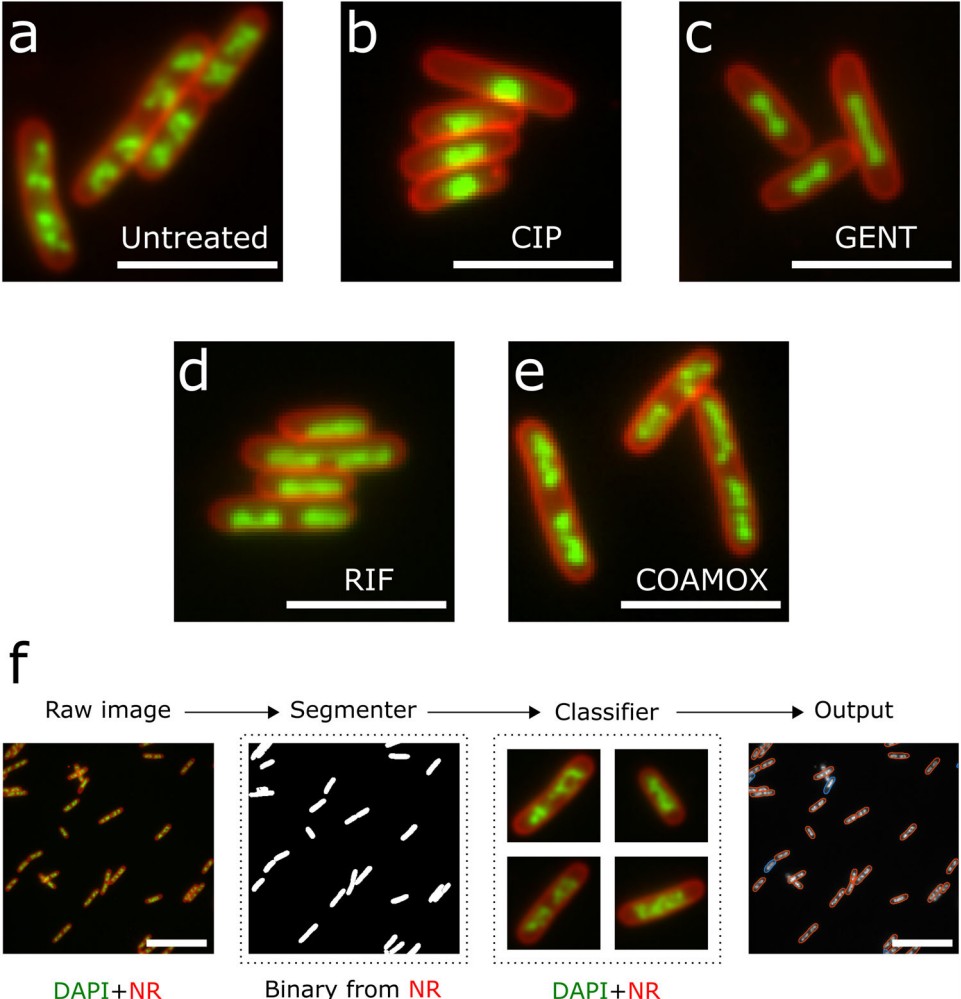

**Fig. 2 Segmentation and Classification pipeline. A** Untreated phenotype in the MG1655 *E.coli* strain, which resembles the resistant phenotype. **B** Ciprofloxacin susceptible phenotype. **C** Gentamicin susceptible phenotype. **D** Rifampicin susceptible phenotype. **E** Co-amoxiclav susceptible phenotype. Note, this phenotype is morphologically similar to the untreated phenotype. **F** A multichannel image consisting of DAPI (green) and Nile Red (red) is split into individual channels. A Mask R-CNN segmenter segments single-instance binary masks from the Nile Red channel. Binary masks are used to isolate single cells. A phenotype classifier classifies individual cells into either the resistant or susceptible phenotype, using both channels. The scalebars in **A**–**E** are 2 μm, and the scalebars in **F** are 5 μm.

DenseNet121 classifier classifies cells into phenotypes using the images generated using both NR and DAPI stains. Together, both stages allow precise quantification of the sample response to antibiotic treatment with single-cell resolution.

**Single-instance cell segmentation by Mask R-CNN.** To segment cells from micrographs, a Mask R-CNN model was built on a ResNet50 backbone, and trained for a 2-class segmentation task (cell/background) on a cross-validation dataset of annotated micrographs of treated and untreated cells from 6 repeat imaging experiments of MG1655 (Supplementary Fig. S4C), consisting in total of 29,297 ground truth, manually curated cells in 459 fields of view. During training, the model was continually validated on a validation set, consisting of 9044 cells in 115 fields of view.

We evaluated the performance of our segmentation on a dataset consisting of all the micrographs across all treatments in a "holdout" experiment – from which no cells or micrographs were used either in training or hyperparameter optimisation. Evaluating a total of 155 micrographs containing 13,247 ground-truth, manually curated cells, we detected a total of 12,147 cells (92 % maximum total recall). To quantify the quality of the

segmentation, we calculated the average precision-recall curves at a range across Intersection over Union (IoU) thresholds, as well as the associated segmentation confidence (Supplementary Fig. S5). The precision-recall curve quantifies the tradeoff between precision (i.e., the fraction of returned results that are relevant), and recall (i.e., the fraction of total relevant results that were successfully returned), whereas the IoU quantifies the area overlap between the detection instances and ground truth instances needed to count as a successful detection – as the IoU threshold increases, the task becomes harder. We achieved a mean Average Precision (mAP) of 70 % at the standard IoU threshold of 0.5 (Supplementary Fig. S6).

**Distinguishing resistant and susceptible single-cell phenotypes.** To classify segmented cells into distinct phenotypes, we trained DenseNet121[38] classifiers in a range of computational experiments (Supplementary Fig. S4B, C), for the binary classification of resistant and susceptible single-cell phenotypes generated using MG1655 and one classifier per antibiotic. As discussed above, treated MG1655 cells were used to generate the susceptible class, whilst untreated cells were used to generate the resistant class.

First, we trained each classifier on the cross-validation dataset, where all untreated and treated cells from 6 experiments were combined into one dataset from which cells for the training, validation and test sets were drawn randomly without replacement (see *Methods*); this "cross-validation" approach provides the expected upper bound for model performance. Second, to examine the performance loss from the upper bound due to experimental variation within the training data, we performed an experimental K-fold cross-validation: one of the 6 experiments was withheld for testing, and the model trained on a class-balanced dataset consisting of an equal number of cells drawn randomly from each of the remaining experiments. This "K-fold cross-validation" experiment was rotated around with the final reported result being the sum over 6 different models trained and tested on different permutations of experiments. Lastly, to evaluate the robustness of the classifiers against experimental variation, we trained a model on a class-balanced dataset of equal numbers of cells drawn randomly from the 6 experiments, and then evaluated it on a class-balanced dataset of the same equal number of randomly selected cells from a 7th, "holdout" experiment. Notably, no data from the holdout experiment was used for training, or hyperparameter optimisation of any of the classifiers.

With this procedure, we achieved an excellent (>84%) single-cell classification accuracy in the holdout experiments across all antibiotic conditions (Fig. 3), with comparable statistics in the other computational experiments; full numerical results across all experiments and antibiotics are provided in Supplementary Tables S2 and S3. Untreated cells, which were used to generate the resistant class, were predominantly classified as resistant (ie. not displaying a physiological response to an antibiotic), whilst treated cells were classified as susceptible (ie. displaying the expected physiological response).

To understand the decision-making process of the CNN classifier, and the potential failure modes, we used saliency mapping[39] to produce attention heatmaps over example single-cell phenotype inputs; such mapping highlights the pixels that contribute most to the classification decision. We observed that, for all antibiotics, in correctly classified cells, the classifier focuses primarily on nucleoid structure and organisation, with some attention given to the membrane, as expected (Supplementary Fig. S7). The same pattern was observed in cells that were misclassified (Supplementary Fig. S8) – such cells did not show the full expected phenotype due to cell-to-cell heterogeneity. For example, some ciprofloxacin susceptible cells did not show a full nucleoid compaction during the treatment window, and were thus classified as resistant.

**Single-cell phenotypic classifications reflect clinical *E. coli* isolate MICs relative to a given treatment concentration for ciprofloxacin.** Having validated both our segmentation and classification models on the MG1655 *E. coli* strain used to generate the training data, we deployed the models on six clinical isolates of *E. coli* (EC1-6), each with a different degree of resistance to ciprofloxacin (as exemplified by the MIC of each isolate, which we measured; see *Methods*) and linked to a resistance genotype derived from sequencing (Supplementary Table S4). The rationale here was to understand whether our approach using short incubation times with high concentrations of antibiotics to obtain resistant/susceptible classifications based on cellular phenotypes meaningfully reflected resistant/susceptible classifications defined by determining minimum inhibitory concentrations (MICs), as used in clinical microbiology. To evaluate the response of the isolates to ciprofloxacin, we used our segmentation model (trained on the cross-validation dataset), and the binary antibiotic

susceptibility classifier (resistant vs susceptible to ciprofloxacin) used to analyse holdout samples as described in the previous section.

Samples of clinical isolates were prepared using two conditions: applying no antibiotic treatment (untreated), or treating cells with ciprofloxacin at the same concentration and duration as the susceptible MG1655 used for training (20× EUCAST[40] break-point concentration [i.e., 10 mg/L], 30 min incubation). These samples underwent the same processing as the training samples, producing collections of micrographs for each sample. Those collections of micrographs were segmented to identify individual cells, and then the classification model was used to classify these cells as either resistant, or susceptible.

Prior to any antibiotic treatment, cells of all clinical isolates were predominantly classified correctly (i.e., in the resistant class – untreated cells showing no response to the antibiotic) and with high confidence; for example, we obtained 77% and 96% resistant classifications for untreated EC2 and EC5 (Fig. 4A, left, and Fig. 4B, left, respectively). Treated cells for clinical isolates with an MIC to ciprofloxacin below the 10 mg/L treatment concentration used in this experiment displayed the susceptible cellular phenotype, and were strongly classified as such, with e.g. 91% of all cells being classified as susceptible in EC2, which had an MIC of 0.03 mg/L (Fig. 4A, right). Treated cells for clinical isolates with an MIC > 10 mg/L were also strongly classified as such, with e.g. 91% of all cells being classified as resistant in EC5, which had an MIC of 108 mg/L (Fig. 4B, right). While it may be possible to improve the accuracy of resistant classifications by ignoring predictions with a prediction confidence below a certain threshold, more data would be required to ascertain the effect of a given prediction threshold, the appropriate value of a prediction threshold, and how the resulting dose-response curve correlates with clinically relevant antibiotic resistant tests.

This pattern was maintained across the library of all six clinical isolates (Fig. 5; see also Supplementary Figs. S9–11 for representative fields of view and overlays showing phenotype detections, and Supplementary Fig. S12 for the total number of cell detections in each repeat of each isolate and treatment condition). In isolates with MICs below the training and treatment concentration (EC1–4), there was a statistically significant (*p*-value < 0.001) increase in the ratio of susceptible classifications based on cellular phenotypes in treated samples, as compared to the untreated samples. In sharp contrast, for resistant isolates with MICs above the training and treatment concentration (EC5-6), there was no statistically significant difference in the ratio of susceptible classifications based on cellular phenotype between treated and untreated samples. The size of the proportion of cells classified as showing a susceptible cellular phenotype appeared correlated with the difference between the isolate MIC and treatment concentration. For example, for isolates with MICs well below the treatment concentration (EC1–3) > 80% of cells were classified as showing a susceptible cellular phenotype in treated samples, whilst for EC4, with an MIC approximating the treatment concentration (MIC = 8 mg/L, treatment concentration = 10 mg/L) only 38% of treated cells were classified as showing a susceptible cellular phenotype. The minimum number of observations required to classify treated susceptible phenotypes with a 90% accuracy and a 1% statistical significance was found to be <400, though significantly less cells are required when the treatment concentration exceeds the MIC by at least an order of magnitude (Supplementary Table S5).

**Changes in single-cell susceptible:resistant classification ratios following incubation with varying ciprofloxacin concentrations provides antimicrobial susceptibility information in 30 min.** After demonstrating that our models can differentiate between clinical isolates resistant and susceptible (based on MIC) to a

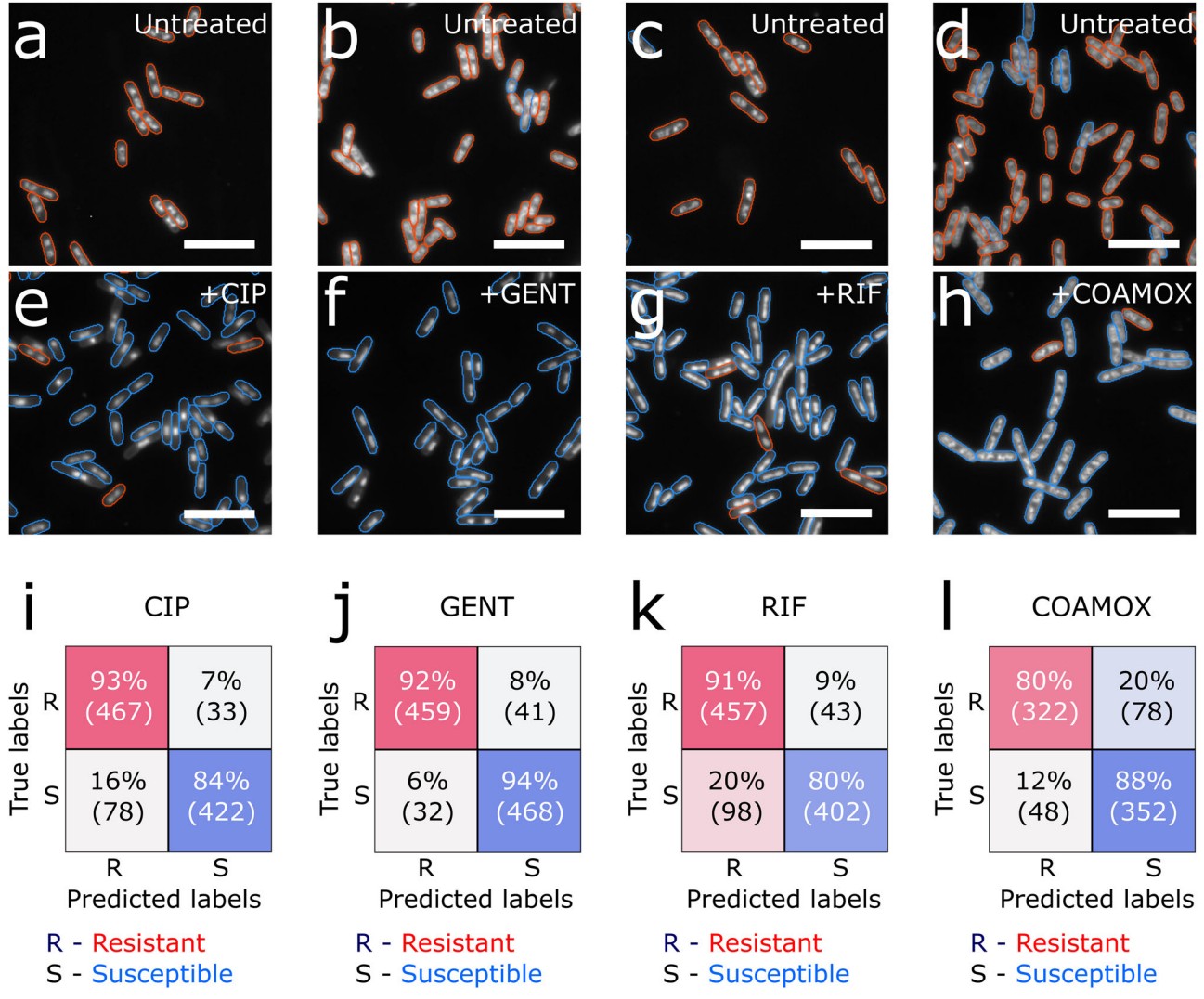

**Fig. 3 Binary classification of resistant and susceptible phenotypes in *E.coli* MG1655 training strain. A** Representative field of view (FoV) of untreated cells shown in grayscale, with an overlay showing phenotype detections. Resistant classifications shown in red, susceptible ones shown in blue. Evaluation carried out with the ciprofloxacin resistant/susceptible classifier. **B** As A, but with the gentamycin classifier. **C** As **A**, but with the rifampicin classifier. **D** As **A**, but with the co-amoxiclav classifier. **E** As **A**, but with ciprofloxacin treated cells. **F** As **B**, but with gentamicin treated cells. **G** As **C**, but with rifampicin treated cells. **H** As **D**, but with co-amoxiclav treated cells. **I** Holdout test performance of the ciprofloxacin classifier, evaluated on class-balanced, randomly sampled 1000 cells from an independent holdout experiment, from which no data was drawn for model training or hyperparameter optimisation. Percentage and absolute class counts shown in both the resistant [R] and susceptible [S] classes. **J** As **I**, but for the gentamicin classifier. **K** As **I**, but for the rifampicin classifier. **L** As **I**, but for the co-amoxiclav classifier, and with a dataset of 800 cells. The scalebars in **A**–**H** are 2 μm.

fixed concentration of ciprofloxacin after 30 min of incubation, we investigated the impact of varying ciprofloxacin concentrations on the ratio of single-cell susceptible:resistant classifications with the same incubation time. The rationale for this was that we reasoned that these ratios may reflect the MIC value for different isolates, based on the observations in Fig. 5. This time, we generated samples of three of the clinical isolates with different MICs (EC1, EC3 and EC5; MICs: 0.008, 0.5 and 8 respectively), treated them at nine different ciprofloxacin concentrations (ranging from 0.001 mg/L to 16 mg/L) for 30 min, imaged them, and evaluated the ratios of cells classified as showing susceptible versus resistant cellular phenotypes (see Supplementary Fig. S13 for numbers of cell detections in each biological replicate of each isolate and treatment concentration).

Across all three clinical isolates, cells treated at sub-MIC concentrations did not show a significant shift away from the untreated/resistant cellular phenotype, and the ratio of cells classified as susceptible:resistant was low. Conversely, at

concentrations in excess of the MIC, a strong response was observed, with >90% of the cells displaying a susceptible cellular phenotype. At intermediate concentrations approximating the MIC for each strain, the magnitude of the response varied logistically between the asymptotes; to quantify this relationship, we fitted asymmetric dose-response models to data (Fig. 6, see also *Methods*).

Our method provided results after as little as 30 min of incubation with ciprofloxacin, whilst gold-standard growth-based ASTs give results over longer periods of time (18–24 h). To compare our result against a growth-based gold-standard, we measured growth curves of the three isolates in the presence of ciprofloxacin over 24 h. From these curves, we calculated the total cell growth by numerical integration of the time-resolved optical density ($OD_{600}$) signal, and normalised it to the growth of untreated cells, thereby creating a ratio of total cell growth as a function of ciprofloxacin concentration, relative to untreated cells (Fig. 6B). Across all three isolates and ciprofloxacin

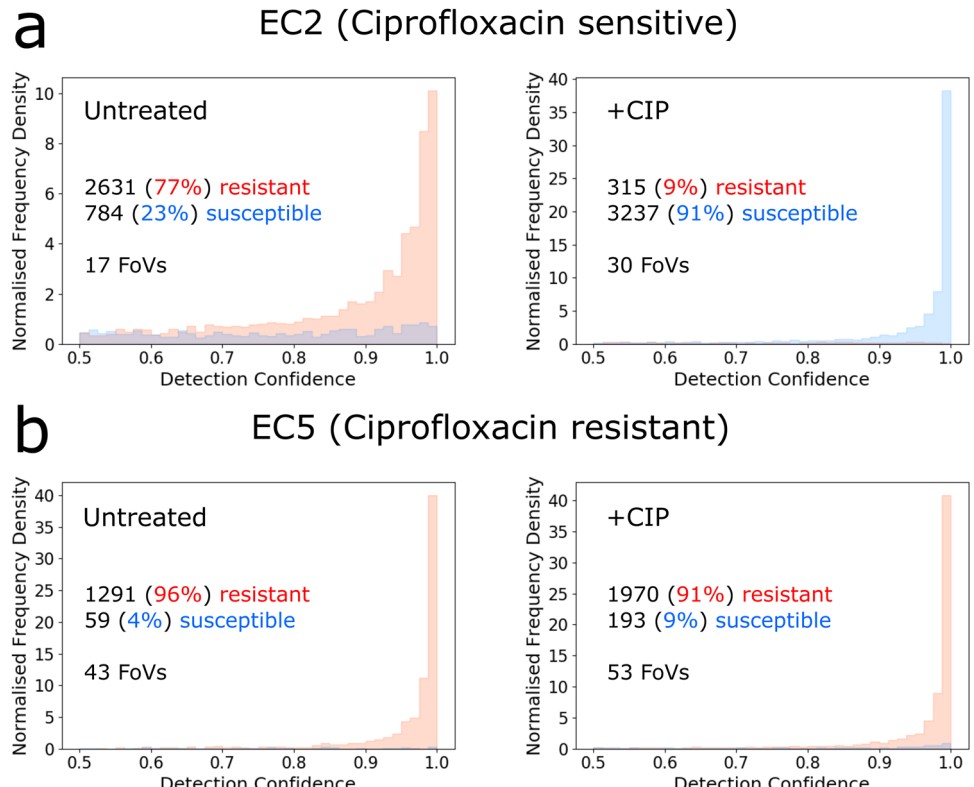

**Fig. 4 Distribution of single-cell classifications of susceptible/resistant reflects the clinical classification of susceptible/resistant, as defined by MIC and EUCAST breakpoints, for two clinical E. coli isolates. A** (left) Histogram of detections from a representative imaging experiment of untreated, susceptible *E. coli* isolate EC2 cells (Clinically ciprofloxacin susceptible, MIC = 0.03 mg/L), normalised to the total sum of detections, as a function of classification confidence. Percentages and absolute cell counts in each class indicated, together with the number of fields of view in the experiment (FoV). (right) As before, but for ciprofloxacin treated EC2 cells. **B** As **A**, but for ciprofloxacin resistant EC5 (Clinically ciprofloxacin resistance, MIC = 72 mg/L).

concentrations, the total cell growth followed a relationship reciprocal to that of the ratio of susceptible:resistant cellular phenotypic classifications derived using our method. At sub-MIC concentrations, cell growth for each strain was not inhibited, leading to a high cell growth ratio, whilst at concentrations in excess of the MIC, no growth occurred. Again, at intermediate concentrations approximating the MIC for each strain, a logistic, dose-response relationship was observed which mirrored the ratio of susceptible:resistant cellular phenotypes we observed using our method.

In both dose-response models, the inflection points of the curves (corresponding to free parameter $c$ in the model fit) were close to the MIC values of the isolates when measured using a routine diagnostic AST assay (i.e. E-tests, see *Methods*). Specifically, for EC1, the measured MIC by E-test was 0.008 mg/L, with a corresponding inflection point at 0.011 ± 0.001 mg/L; further, for EC2, the measured MIC by E-test was 0.5 mg/L, with curve inflection at 0.49 ± 0.18 mg/L. Our results provide evidence that for ciprofloxacin our approach can provide valuable MIC-related information for different clinical isolates following antibiotic incubation times as little as 30 min.

**Discussion**

By combining fluorescence imaging and deep-learning, we demonstrate our DASP platform provides a proof-of-principle approach to rapid single-cell antibiotic susceptibility profiling for *E. coli* across a range of antibiotics with different mechanisms of action. DASP is compatible with analysing clinical isolates with varying susceptibilities to clinically relevant antibiotics such as

ciprofloxacin, providing equivalent information to a traditional growth-based assay in 30 min. MIC-level predictions can be generated by modelling dose-response effects on susceptible:resistant cellular phenotype ratios across an antibiotic dilution series similar to that used in clinical microbiological AST assays.

The high accuracy statistics across our experiments establish that our method is robust against both experimental variability and phenotype plasticity exhibited by different clinical *E. coli* isolates, showing promise for further work evaluating additional antibiotics and moving towards the development of a clinical diagnostic assay. Considering that distinct nucleoid and membrane phenotypes have been identified a wide range of treatments and organisms;[32–34] our method of susceptibility phenotyping should be applicable to a wide variety of antimicrobials and bacterial species, although this remains to be evaluated.

Deploying our models on clinical isolates shows that the models can distinguish between resistant and susceptible isolates around a fixed treatment point, which could facilitate their use for decision-making around standard clinical breakpoints used for resistance classification, such as EUCAST breakpoints. Our comparison with gold-standard bacterial growth assays in the presence of a dilution series of ciprofloxacin also established that our approach can provide equivalent information regarding the ciprofloxacin MIC of a certain isolate in as little as 30 min, limited only by the physiological response rate of the bacteria themselves, and is therefore well-suited to the ultimate goal of a rapid susceptibility testing assay.

Our assay may also offer a route towards a more detailed clinical definition of the MIC value, which is currently only defined by growth. Notably, the MIC value is emerging as an important independent factor in clinical management; e.g., co-

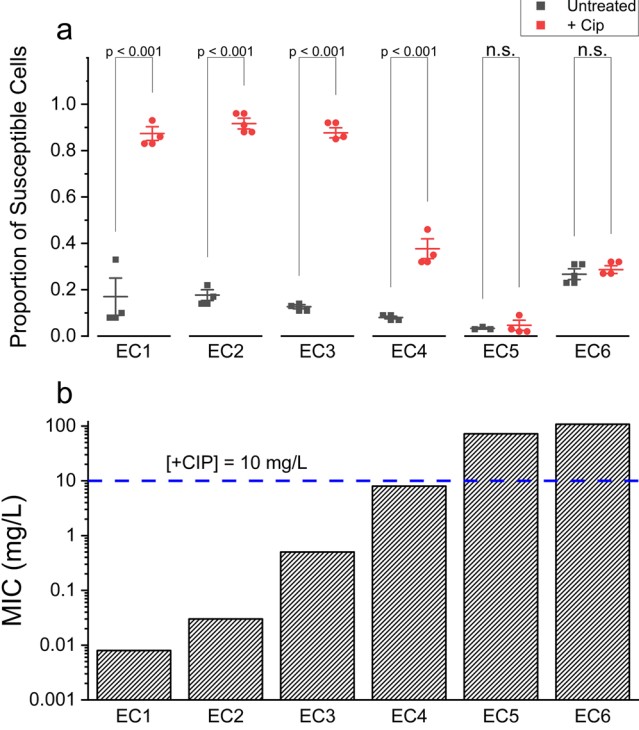

**Fig. 5 Changes in detection distribution upon treatment correlate with the degree of resistance in clinical isolates. A** Ratio of susceptible phenotype detections across different clinical isolates, in untreated and ciprofloxacin treated (+CIP) samples. Alongside the raw data, the mean and standard error of the mean are shown for the 3 biological replicates of each clinical isolate. Overlay shows Tukey range test *p*-values, carried out pairwise between corresponding untreated and treated sets of repeats (*n* = 3), test carried out at significance level of 0.05, "n.s." indicates not significant. **B** Ciprofloxacin MICs of the clinical isolates, derived experimentally as described in Methods. Horizontal line indicates the treatment concentration used in the treated samples, which was used for both clinical isolates and MG1655 training data.

amoxiclav MICs >32 mg/L have recently been specifically associated with poor outcomes in *E. coli* bloodstream infections[41]. Better and faster definition of bacterial MICs may therefore be relevant to optimising treatment strategies and outcomes for individual patients.

**Comparison with other assays**. Compared to current assays, our technique could also serve as a richer potential source of clinical information. Currently established ASTs operating at the colony level only offer aggregate, sample-wide information measured through secondary markers correlated with resistance, such as culture growth or genomic information. This might miss relevant mixed bacterial populations or important heteroresistance phenotypes[42] which could be picked up by evaluating individual cellular susceptibility phenotypes using our method. Compared to other candidate single-cell ASTs, DASP offers the potential to be faster by allowing simultaneous interrogation of large numbers of single-cells (in contrast to previous cytometry or Raman approaches, which only interrogate one cell at a time), and does not require cell tracking (in contrast to previous widefield microscopy approaches focusing on single-cell growth or motion). As this method has only been demonstrated for *E. Coli*, with a relatively small number of antibiotics, significantly more work will be required to establish this method as a reliable and clinically relevant AST.

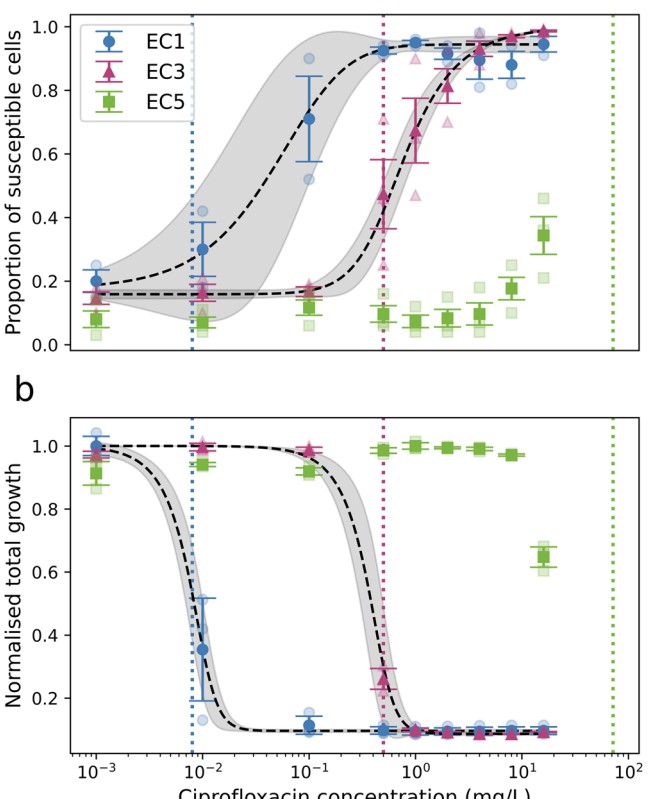

**Fig. 6 Single-cell phenotyping provides equivalent information to a 24 h growth assay, in 30 min. A** Ratio of susceptible detections as a function of ciprofloxacin treatment concentration. Average of 3 biological replicates, error bars show the standard error of the mean. Vertical coloured lines show the ciprofloxacin MICs of the isolates used. Black lines show the dose-response fit, grey regions show the 95% confidence band of the fit. **B** Total bacterial growth in liquid culture over 24 h, in the presence of ciprofloxacin, normalised to the growth of untreated cells. Measured MICs of the isolates are: EC1 – 0.008 mg/L, EC3 – 0.5 mg/L, EC5 – 72 mg/L.

Previous single-cell phenotypic studies have implemented linear transformations and manual analysis of engineered features to classify methicillin susceptible and resistant *S. aureus* cells[31]. Our approach represents a step-change over the approach above, since it transforms the assessment of antibiotic susceptibility into a non-linear classification of resistant phenotypes against susceptible phenotypes, addressed by CNNs. Using learnt features as opposed to engineered ones is advantageous, since it generalizes the technique by allowing subtle phenotypic changes to be detected (as seen in our co-amoxiclav results), and reducing human bias. In contrast to a more manual analysis, CNNs offer automatic processing of large volumes of data, scaling better with the aim of a rapid, robust assay.

**Future extensions**. Our approach could be extended to become more scalable, and to avoid the need for pre-culture steps. Currently, the assay operates on cultured clinical isolates, and thus does not yet reduce the time needed to isolate and grow micro-organisms from patient specimens. The isolates are cultured to a constant OD$_{600}$ of ~0.2 prior to processing, translating to ~10$^8$ Colony Forming Unit (CFU) counts - a count much higher than the CFU counts encountered in infected physiological body fluids[43]. We envision that use of microfluidics will be instrumental in bypassing the pathogen isolation and culture steps by isolating and concentrating bacteria from patient specimens.

Finally, our assay has the potential to be coupled with rapid bacterial species identification which can be performed using various methods, such as targeted FISH staining[25,44].

**Limitations**. There are several limitations to our study. Our deep-learning approach relies on explicit classification of phenotypes; whilst that removes the need for engineered features, enables high-throughput and reduces human bias, it still requires that phenotype classes are homogenous across cells and isolates. Further work is required to validate this approach with more species and antibiotics, and to more robustly demonstrate that cellular susceptible:resistant phenotypes derived from short periods of antibiotic incubation correlate with currently used clinical microbiological susceptibility metrics such as MIC. Our approach also requires specific models to be trained for each combination of antibiotic and species, which may not scale well with the size of the problem space; however, a reformulation of the computational task should produce solutions that scales better.

## Methods

**Bacterial strains and sample preparation**. The reference laboratory strain was *E. coli* MG1655. Clinical isolates were blood culture isolates of *E. coli* processed for diagnostic purposes and stored by the Microbiology Laboratory of the Oxford University Hospitals NHS Foundation Trust, Oxford, UK. Individual colonies of *E. coli* were cultured overnight in 5 ml of Lysogeny Broth (LB) at 37 °C, then diluted 1:100 by volume in 5 ml of EZ Rich Defined Medium (RDM; Teknova) and cultured at 37 °C until reaching an $OD_{600}$ of ~0.2 in a shaking incubator. Subsequently, 1 ml aliquots of the culture were treated with one of the antibiotics (Supplementary Table S1) at the concentration and duration listed to produce cells showing the antibiotic-specific susceptible phenotype; aliquots of the culture were treated similarly but in the absence of an antibiotic in order to produce the resistant phenotype. As antibiotic concentrations at EUCAST breakpoints produced less distinct cellular susceptibility phenotypes within 30 min (e.g. Supplementary Fig. S14 for ciprofloxacin), for model training purposes the treatment concentrations were optimised to produce distinct single cell phenotypes for each antibiotic and were well in excess of the EUCAST breakpoint. Cells were then fixed by incubation in 2.5% formaldehyde solution for 30 min. Cells were washed 3 times with PBS via centrifugation at 4500 RCF for 3 min, and then incubated with 100% ethanol for 10 min. Cells were resuspended in PBS and stained by adding 10 µg/L DAPI as the nucleic acid stain (GeneTex, catalogue number GTX16206) and 1 µg/L Nile Red as the membrane stain (Fisher Scientific, catalogue number 10464311) and incubated at room temperature for 10 min. Cells were washed twice with PBS and suspended in a small volume of 10–20 µl.

For ciprofloxacin titration assays, the antibiotic was diluted across a concentration range from 16 mg/L to 0.001 mg/L and co-incubated with 1 ml aliquots of bacteria as described above. Stained and prepared samples were imaged by mounting on agarose pads. Agarose pads were prepared consisting of 1% high-purity agarose (Bio-Rad, catalogue number #1613101) in half-concentration PBS solution, and imaged inverted through a glass slide that had been burned in a plasma cleaner at 500 °C for 60 min.

All clinical isolates in the study had been whole-genome-sequenced on the Illumina platform as described previously[45], and AMR genotypes were assigned using the ResFinder[46] database with Abricate v0.9.8[47] (--min-id 95 --min-cov 95). The MICs of the clinical isolates were calculated empirically by E-test strip (Lioflchem), or where the MIC exceeded the maximum range of the strip, by a 1:1.5 broth dilution of the antibiotic.

**Bacterial growth curves**. Individual colonies of each strain were grown overnight at 37 °C in LB broth and subsequently diluted to $OD_{600}$ of ~0.04 (1:100 dilution) in RDM. These cells were added in equal volume to a microtiter plate containing a prepared 2x dilution range of ciprofloxacin in RDM to a final volume of 200 µl. Inoculated plates were incubated at 37 °C in a Tecan Sunrise plate reader, with an $OD_{600}$ reading recorded at 15-min intervals, following a 5-s orbital shaking. The same measurement was taken for a blank sample, consisting solely of the growth medium. To calculate total cell growth (Fig. 6B), the time-resolved $OD_{600}$ signals were integrated numerically in time. From this, the integrated blank signals were subtracted, and finally all measurements were normalised to the growth of untreated cells by dividing each measurement by the measurement coming from untreated cells.

**Imaging**. Agarose-mounted samples were imaged on a Nanoimager-S fluorescence microscope (Oxford Nanoimaging). Briefly, a blue (405 nm) and a green (532 nm) laser were combined using a dichroic mirror and coupled into a fibre optic cable. The fibre output was focused into the back focal plane of the objective (100× oil immersion, NA 1.4) Fluorescence emission was collected by the objective, separated into two emission channels and imaged onto a sCMOS camera (Orca flash V4, Hamamatsu). To make best use of the camera dynamic range DAPI signal was imaged using 405 nm excitation and Nile Red signal was imaged using 532 nm excitation; both signals were acquired consecutively. To ensure reproducibility, laser powers were kept constant at 1.5 kW/m². For each of the two channels, for each field of view (FoV), a stack of 30 frames was acquired at 30 ms exposure and 33 Hz frequency. To automate the task and reduce human bias, the multiple acquisition capability of the microscope was used, and the microscope autofocused on each FoV prior to acquisition.

**Deep learning – model selection**. The segmentation and classification models employed in this study were chosen by evaluating a range of classification models (including models from the DenseNet, ResNet and EfficientNet family) and state of the art segmentation models including Mask R-CNN[37], Cellpose[48] and YOLO v8[49] (Supplementary Figs. S15 and S16). The batch size and learning rate of each model was found using a grid search, and the best pipeline was ultimately found by evaluating the classification accuracy on the holdout test set. The best combination of models was found to be Mask R-CNN and DenseNet121[50]. A breakdown of the classification accuracies for each model and phenotype are shown in Supplementary Table S6.

**Deep learning – segmentation**. To generate training data for the Mask R-CNN segmenter, only the Nile Red channel of every FoV acquired was used. The 30 frames of each FoV were averaged to generate a grayscale image, which was further expanded to RGB space by replicating the grayscale image in each colour channel. The raw images were augmented on-the-fly by random cropping to a size of 256 by 256 pixels, followed by a random sequence of transformations including horizontal and vertical flips and translations, rotations, cutout[51] as well as Gaussian blurring. Such augmented images were passed forward to the segmenter during training, along with equivalently transformed ground-truth instance segmentation masks; these segmentation masks were generated by manual data annotation followed by boot-strapping and manual curation. The internal parameters of Mask R-CNN were optimised to match the task at hand, consisting of modifications to its Region Proposal Network (RPN) and Non-Maximum Suppression parameters. The segmenter training was

then optimised via a grid-search, keeping the model that performed best on validation data.

In the end, the best performing model was trained using an initial learning rate of 0.003 and batch size of 2 with the Adam[52] optimiser, using a momentum of 0.9 and weight decay of 0.001. The final model was trained in 4 consecutive steps, starting from initial weights trained on the MS COCO dataset[53]. In the first step, the model 'top', consisting of the RPN and a second stage classifier and mask regressor were trained for 50 epochs at the initial learning rate, with other weights frozen. In the second step, the entire network was trained together, including the feature-encoding backbone, at the initial learning rate for another 50 epochs. In the 3rd step, the 'top' was fine-tuned for another 50 epochs, this time using 10 % of the initial learning rate (0.0003). Finally, the entire network was fine-tuned at 10 % of the initial learning rate for the final 50 epochs.

The Mask R-CNN model was adapted from a standard implementation[54].

**Deep learning – classification**. To generate training data for the DenseNet121 classifier, both channels of every FoV acquired were used. The 30 frames of each FoV were averaged separately for both channels and used to construct RGB images, with Nile Red signal in the red channel and DAPI signal in the green. The DAPI channel was registered automatically to the Nile Red channel using cross-correlation[55] to correct for any drift between the channels. Individual cells were extracted from assembled images using the ground-truth instance segmentation masks that were used to train the segmenter. All cells were then resized to a common size of 64 by 64 pixels by zero-padding in either dimension if below the target size, or resized down to target size if above. To compensate for differences in staining and illumination, histogram equalisation was applied to every cell, independently for each channel, within the segmentation mask only. Cells were then augmented on-the-fly using a random sequence of affine transformations, followed by a random sequence of intensity augmentations to increase robustness against experimental variation – these include a unsharp masking, random brightness modifier in HSB colour space, addition of Gaussian-distributed noise, channel misalignment and random Gaussian blurring (Supplementary Fig. S17). The classifier training was then optimised via grid-search, keeping the model that achieved the best accuracy on validation data. The training loss and accuracy for each model are shown in Supplementary Fig. S18.

To train the model to recognise binary resistant/susceptible cellular phenotypes, individual untreated and treated MG1655 susceptible cells only were used.

The classifier was implemented in Keras[56] version 2.2.4.

**Deep learning – saliency mapping**. To produce attention heatmaps over example classification inputs, we calculated the gradient of the output category with respect to the input single-cell image. We propagated positive gradients for positive activations only[39], and visualised the absolute value of the gradient.

**Segmentation metrics**. The quality of Mask R-CNN segmentation was analysed using Precision-Recall curves using the bounding boxes of detections and ground truth segmentations to compare performance at various IoU thresholds.

**Classification metrics**. Classification metrics of binary resistant-susceptible classifiers are presented as a confusion matrix, which displays the True Positive (TP), True Negative (TN), False Positive (FP) and False Negative (FN) counts in each class. From these, per class precision, recall and accuracy can be calculated as

follows:

$$Precision = \frac{TP}{TP + FP} \tag{1}$$

$$Recall = \frac{TP}{TP + FN} \tag{2}$$

$$Accuracy = \frac{TP + TN}{TP + TN + FP + FN} \tag{3}$$

**Dose-response model fitting**. To model the ratio of cells classified as susceptible as a function of treatment concentration, we use non-linear least squares to fit a generalised logistic function of the following form:

$$f(x) = d + \frac{a - d}{\left(1 + \left(\frac{x}{c}\right)^b\right)^g} \tag{4}$$

where $a$ and $d$ are the lower and upper asymptotes, $b$ is the scale parameter, $c$ is the x-coordinate of the inflexion point, and $g$ is the asymmetry parameter. The confidence bands (CB) of the fit can be calculated directly from the covariance matrix of the fit:

$$CB(x) = y(x) \pm t_{\frac{\alpha}{2}, \nu} \sqrt{\chi_\nu^2 \sum_{j,k=0}^{n} \frac{\partial f(x)}{\partial p_j} \frac{\partial f(x)}{\partial p_k} C_{jk}} \tag{5}$$

where $y(\hat{x})$ is the best-fit estimate at $x$, $t_{\alpha/2, \nu}$ is the upper $\alpha/2$ critical value for the t-distribution with N-n degrees of freedom, $\nu$ is the degrees of freedom, $\chi_\nu^2$ is the reduced chi-square of the fit, $C$ is the covariance matrix, $p$ are the best-fit parameters and $f(x)$ is the generalised logistic function.

**Statistics and reproducibility**. The ratios of susceptible cells found and classified in untreated and ciprofloxacin treated clinical isolates were analysed using Tukey's range test, computed pairwise for untreated and treated samples of each clinical isolate, and separately for each isolate. Three biological repeats were analysed for each isolate, producing a set of three susceptible cell ratios ($n = 3$) per isolate, per biological repeat. The total number of cells that contributed to each ratio is displayed in Supplementary Fig. S13. The significance level of the test was set at 0.05.

**Ethics**. We analysed bacterial isolates routinely stored by the John Radcliffe Hospital Microbiology laboratory; no sampling was specifically undertaken for the purposes of this study and no patient-related information was accessed. In the UK, bacterial isolates routinely cultured from human clinical samples do not require ethical approval for analysis under the provisions of the Human Tissue Act as they do not contain any material considered to be human tissue.

**Reporting summary**. Further information on research design is available in the Nature Portfolio Reporting Summary linked to this article.

## Data availability
The data and model weights that support the findings of this study, are available from the Oxford University Research Archive: https://ora.ox.ac.uk/objects/uuid:12153432-e8b3-4398-a395-abfb980bd84e. The source data for Figs. 5, 6, S15, S16 can be obtained in supplementary dataset 1–4, respectively.

## Code availability
All code used to generate the results is available publicly on the Kapanidis Laboratory GitHub account, accessible at: https://github.com/KapanidisLab/Deep-Learning-and-

Single-Cell-Phenotyping-for-Rapid-Antimicrobial-Susceptibility-Testing. Detailed guidance on using the code is available upon request from AZ.

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

## Acknowledgements

This work was supported by an Oxford Martin School (by the establishment of the Oxford Martin School Programme on Antimicrobial Resistance Testing; to A.N.K., N.S., C.N., D.C. and M.A.), by Wellcome Trust grant 110164/Z/15/Z (to A.N.K.) and by UK Biotechnology and Biological Sciences Research Council grants BB/N018656/1 and BB/S008896/1 (to A.N.K.). The work was also supported by an UK Engineering and Physical Sciences Research Council funded scholarship (to A.Z.), part of the Oxford-Nottingham Biomedical Imaging Centre for Doctoral Training grant EP/L016052/1. The research was additionally supported by the National Institute for Health Research (NIHR) Health Protection Research Unit in Healthcare Associated Infections and Antimicrobial Resistance (NIHR200915) at the University of Oxford in partnership with United Kingdom Health Security Agency (UKHSA) and by the NIHR Oxford Biomedical Research Centre. N.S. is an NIHR Oxford BRC Senior Research Fellow. The views expressed in this publication are those of the authors and not necessarily those of the NHS, NIHR, the Department of Health or Public Health England. The authors would like to acknowledge and thank Mantas Krisciunas whose early exploratory work provided the foundation of this project.

## Author contributions

A.K. conceived and supervised the project. A.Z., H.E.S. and A.N.K. designed experiments. A.Z. carried out experiments, developed, trained and optimised computational models, analysed data. P.T. trained the models to evaluate the most optimum deep learning pipeline and segmented the microscopy images used for training the models. A.Z. and P.T. both contributed equally to this work. C.F. and H.E.S. assisted with experiments. L.B. and S.O. provided clinical isolates used in this work. C.N., N.S., M.A., H.E.S., M.S., D.C. and A.N.K. contributed to the interpretation of the results and shaped the strategic direction of the project. AZ and ANK wrote the manuscript. All authors discussed the results and contributed to the final manuscript.

## Competing interests

The work was carried out using a wide-field microscope from Oxford Nanoimaging, a company in which A.N.K. is a co-founder and shareholder. The other authors declare no competing interests.
