## [Peer Review File · Communications Biology]

Reviewers' comments:

Reviewer #1 (Remarks to the Author):

This paper by Zagajewski et al. proposes a method for rapid phenotyping of bacterial susceptibility to various antibiotics using fluorescence-based imaging of membrane/nucleoid staining and deep learning. The method does not require cell growth and can therefore provide timely information about the presence of antibiotic-resistant cells in a population. The study shows convincingly that the method works overall with proper considerations with regards to statistical analyses. The wide range of antibiotics used and corresponding phenotypes highlight the versatility of the technique. Its application to clinical isolates and inference of MIC values further show potential for translation into clinics.

The combination of single-cell segmentation (Mask-R-CNN) and classification (DenseNet) makes the analysis pipeline fairly complex. The segmenter has been trained using close to 30k cells from >450 images. Is there evidence that so many images are necessary for finding cells with such highly reproducible appearance? Generally, could one design a single object detector/segmenter (e.g. YOLO, SSD-based) to perform both identification and classification at the same time?

The classifier itself has a high mAP, but will be ultimately limited by human labelling errors in the presence of unclear phenotypes. Could one design an additional control to confirm correct classification of the ambiguous phenotypes when building the training set?

Figure 4 shows that cells with low detection confidence (e.g. <0.7) should not be used to draw conclusions as the untreated isolate cells have similar amounts of susceptible/resistant phenotypes when they should be all classified as 'resistant'. This could be highlighted in the discussion.

Minor comments:

-in Figure 1, the y-axis for cell count shown goes only up to 15 but presumably a typical histogram would have orders of magnitude more cells.

-What is a typical proportion of cells not identified by the Mask-R-CNN segmenter? It is clear from Figure S11 for instance that many cells are not found.

Reviewer #2 (Remarks to the Author):

This manuscript proposes a new method for rapid antimicrobial susceptibility testing (AST) using deep learning and single-cell phenotyping. The proposed approach first employs single-cell phenotyping to phenotype the target cells. Next, Mask R-CNN and Desnet121 are utilized to segment the required cells and classify them into distinct categories. Extensive experiments were conducted to verify the efficacy of the proposed method, which was found to be more efficient than current gold-standard AST. The idea of using deep learning methods for microscopy-based AST is very interesting. However, the manuscript has several weaknesses that are worth noting:

1. The performance evaluation in the manuscript needs to be further specified and improved. For instance, in the context of segmentation, more evaluation metrics, such as mean intersection over union, are required to better demonstrate the performance of Mask R-CNN. The authors may refer to a review article titled "Deep Learning for Imaging and Detection of Microorganisms" to select appropriate models for comparison. Additionally, the authors should ensure that vocabulary usage is consistent throughout the manuscript. It would be better to use "precision" and "recall" in the definition of "sensitivity" and "specificity".
2. The proposed methods in the manuscript lack comparison with other existing methods. Although the authors present the results of Mask R-CNN and DenseNet121, no comparisons with other deep

learning models are provided. Therefore, it is difficult to determine whether the performance of the proposed methods is the best. To address this issue, the authors should include comparisons with other models, such as Swim-Transformer and U-Net, to better assess the effectiveness of their proposed methods.

3. In the manuscript, the authors evaluate the robustness of DenseNet121 by training and testing on two class-balanced datasets. However, this approach may not be sufficient to validate the robustness of the proposed method. A better way to evaluate the model's robustness would be to perturbate the input images, such as by adding random noise or cropping, to simulate the variations that occur in real-world applications. The authors should report the changes in performance resulting from such perturbations. If the model can maintain a high per-class accuracy despite these perturbations, it would demonstrate its robustness.

4. The proposed models in the manuscript require further improvement. Firstly, it is worth considering whether it is necessary to use DenseNet121 for classification. Why not use Mask R-CNN to perform segmentation and classification simultaneously, which could potentially improve efficiency? Secondly, the models of Mask R-CNN and DenseNet121 are already widely used in various fields. Therefore, the authors should demonstrate why they choose those two models over other deep learning methods, either by providing a comparison with alternative approaches or by highlighting the unique value of their proposed method in this manuscript.

5. The manuscript contains several careless mistakes. For instance, in the Dose-Response Model fitting section, the authors neglected to italicize α in the phrase "upper $\alpha/2$ critical value." Additionally, citations 38 and 39 are from the same proceedings but have different names. To improve the manuscript's reliability, the authors should carefully review its details to ensure the accuracy of its contents.

6. To aid readers in understanding the scale of different microscopic images of bacteria, it would be helpful for the authors to include scale bars for the following figures: Fig. 2, 3, S1, S3, S4, S8, S9, S11, S12, and S13.

7. To ensure reproducibility, the images used for deep learning training and testing should be made publicly available. I strongly recommend that the authors upload their datasets to an appropriate public data repository. This will allow other researchers to evaluate the performance of the proposed method and compare it with other methods on the same dataset. Furthermore, it will enhance the transparency and credibility of this work, which is essential for the community.

8. Statistical analysis should be performed on Fig S7 by adding standard deviations for sensitivity, specificity and accuracy. This can be achieved by K-fold cross-validation or independent repeating experiments. Additionally, it's worth noting that Figures S7 and S10 are tables, not figures, so they should be referred as Tables accordingly.

9. The authors should consider providing some guidances for future studies. This would be particularly useful for those who want to employ this model for analytical study or high throughput drug screening.

Reviewer #3 (Remarks to the Author):

The paper describes a novel deep learning-based, optical imaging technology applied to looking at antimicrobial susceptibility testing (AST) in E.coli isolates. The technique is interesting and the process for training the system on images generated are appropriate and informative. Although the data generated is of interest to researchers in the field, I feel the paper overstates its current impact on the development of ASTs for a number of reasons as outlined below.

I feel the title overstates the findings of the paper. The data generated is only for E.coli and a significant proportion of the data generated relates to lab reference strain MG1655. I would suggest this needs to be acknowledged in the title (so include in E.coli) as a minimum, but this perhaps needs to go further to say that what is being measured is response(s) to antibiotics, not informing ASTs at this stage and with no evidence that it would work with other bacteria.

The authors refer to other methods being low throughput in the abstract, but make no reference to the likely throughput of the imaging methods described. The time savings are also misleading as they only relate to the time taken to generate antibiotic susceptibility data, not acknowledging that there is a prior culture phase before this.

The paper describes responses of lab reference strains MG1655 to 4 antibiotics, with different modes of action. The authors state that rifampicin is clinically relevant but this is not true for E.coli, noting that in the supplementary section they acknowledge that no breakpoint has been determined for rifampicin in E.coli. I would suggest this needs to be made more clear and ideally another antibiotic selected. The authors also don't make it clear in the text of the results section that the concentrations of antibiotic used for the training experiments are 20-fold higher than the concentration which is used to define clinical resistance. The authors also don't state (as far as I can see) the MICs that were determined for MG1655 with each of these antibiotics and, as such the concentration used may be very high compared to the MIC and to the MICs which you might expect to see with clinical E.coli strains. If the paper is about ASTs then this appears to be completely counterintuitive to me. The authors need to make it very clear why they use such high concentrations and not the resistance breakpoint MIC, as others have done.

Given that the authors only test 4 antibiotics, it's perhaps concerning that the antibiotic class used most commonly in the NHS (represented here by Co-amoxiclav) only gives a very subtle phenotype, despite the very high concentrations of antibiotic used relative to breakpoint. If the message of the paper is that this is a new rapid AST platform, then this observation would appear to make it unsustainable and needs to be addressed by the authors. This is before any evaluation of other beta lactam antibiotics and/or clinical isolates for this class of antibiotics. This is a major concern about the validity of the overall approach.

The authors talk about whole population analysis and this is clearly one of the strengths of the approach. They don't comment on the proportion of the cell population which would need to show a resistant phenotype before the isolate could be classed as resistant, although this is clearly evident from the proportions of the populations that respond to different concentrations of ciprofloxacin (only) in the MIC experiments. This also assumes that all of the strains tested are homo-resistant to ciprofloxacin, which might not be the case. Again this needs to be addressed strongly in the paper, but could be an important feature that advocates for the use of this technology.

The statement that the inflection point of the analysis for the MIC determined by the optical imaging is close to the clinical AST determined value is perhaps misleading, given that one is a numeric value and the MIC is based on a log₂ dilutions series. There is no detail of the AST method used, otherwise it might be appropriate to also determine the inflection point from plotted optical densities in broth dilution assays rather than as for standard assays.

Overall my view is that this is an exciting technology but there are significant gaps in the data provided that do not support its claims to be a potential platform for rapid ASTs. I would suggest considerable revisions of the manuscript to either address these points or to make it clear that this is currently an experimental method only.

DEPARTMENT OF PHYSICS
KAVLI INSTITUTE FOR NANOSCIENCE DISCOVERY
Sherrington Road, Oxford, OX1 3QU, UK
Tel: +44 (0)1865 272 226
Email: kapanidis@physics.ox.ac.uk

25th July 2023

Dear Reviewers,

We are grateful for the time you have taken in reviewing our manuscript COMMSBIO-23-0368-T with the revised title "*Deep Learning and Single-Cell Phenotyping for Rapid Antimicrobial Susceptibility Detection in Escherichia coli*" (former title: "*Deep Learning and Single Cell Phenotyping for Rapid Antimicrobial Susceptibility Testing*").

We appreciate the detailed feedback and constructive comments provided, which have significantly contributed to improving our work. Most notably, we have further justified the models employed in our deep learning pipeline by additionally exploring 12 classification and 3 segmentation models, establishing Mask-RCNN and DenseNet121 were indeed the best choice of model. In addition, we have changed the title and emphasised the proof-of-concept nature of our work in detecting antimicrobial resistance; future work will focus on clinical application. Our pilot work does, however, clearly demonstrate that in our evaluations of the impact of ciprofloxacin on clinical *E. coli* isolates, incubation with high antibiotic concentrations over 30 minutes induces cellular changes detected by our models that correlate very well with susceptible/resistant classifications derived using standard microbiological methods (i.e. inhibition of growth at a clinical breakpoint established by The European Committee on Antimicrobial Susceptibility Testing (EUCAST)); these results serve as a clear testament to the promise of our new method.

We have considered each point raised by the reviewers and have addressed them with changes to the text and figures, resulting in additional figures and tables in the supplementary information. Some of these figures are appended to the end of this document to aid addressing the reviewers' comments. In this letter, we also include a point-by-point response to all reviewers.

We believe that our revised manuscript is currently suitable for publication in *Communications Biology*, and we are looking forward to hearing from you.

Many thanks for your kind consideration.

Sincerely,

Achillefs Kapanidis, Ph.D.

Professor in Biological Physics

University of Oxford, UK

Reviewer 1

The combination of single-cell segmentation (Mask R-CNN) and classification (DenseNet) makes the analysis pipeline fairly complex. Generally, could one design a single object detector/segmenter (e.g. YOLO, SSD-based) to perform both identification and classification at the same time?

- We have trained binary classifiers for both Mask R-CNN and YOLO v8 for all phenotypes featured in this paper and found that they do not outperform the models we chose for this paper (Mask R-CNN and DenseNet121); the mean test accuracy across all four antibiotic phenotypes for DenseNet121, YOLO v8 and Mask R-CNN are $87 \pm 4 \%$, $71 \pm 11 \%$ and $78 \pm 22 \%$, respectively (Fig. R1; incorporated in the revised ms as Fig. S15). The raw classification data were added to the ms as Table. S5.

The segmenter has been trained using close to 30k cells from >450 images. Is there evidence that so many images are necessary for finding cells with such highly reproducible appearance?

- For more distinct cellular phenotypes (such as those induced by exposure to ciprofloxacin), fewer images would be required to reach respectable accuracies, but for less distinct phenotypes (such as those induced by co-amoxiclav and gentamicin), significantly more images may be required. This is evidenced by the variation in the K-fold cross-validation accuracy for each antibiotic (Fig. R2; added to the ms as Table S3).

The classifier itself has a high mAP, but will be ultimately limited by human labelling errors in the presence of unclear phenotypes. Could one design an additional control to confirm correct classification of the ambiguous phenotypes when building the training set?

- For each experiment, each bacterial sample was treated separately with a different antibiotic. As such, in each image, the antibiotic label was known, and labelling errors would not occur. Since there are natural variations in the cellular phenotypes within and between experiments, we made the model more robust by training it on data from multiple experiments. To ensure there is a full and consistent conversion to a ciprofloxacin phenotype, longer incubation times could be employed. However, this may increase the false negative rate, as the models will be less sensitive to more subtle/transitioning ciprofloxacin phenotypes.
- Further, each model was a binary classifier, so for each antibiotic the model would predict whether the cells presented to it were affected by antibiotic treatment (i.e. susceptible) or not (i.e. resistant). Binary classifiers were chosen over a single multi-class classifier, as this would make the most sense in the clinic where clinical susceptibility is evaluated per antibiotic, conventionally by using a growth-based assay defining the minimum inhibitory concentration (MIC) of an antibiotic for a bacterial culture.

Figure 4 shows that cells with low detection confidence (e.g. <0.7) should not be used to draw conclusions as the untreated isolate cells have similar amounts of susceptible/resistant phenotypes when they should be all classified as 'resistant'. This could be highlighted in the discussion.

- Our intention in Figure 4 was to show the variation in prediction confidence for two clinical isolates with MICs below and above the clinical breakpoint. We have now clarified this in the Legend and Figure labels. While it would be possible to remove predictions of low confidence to improve the prediction of resistant cells, more data would be required to ascertain the appropriate value and effect of a prediction threshold, and to determine how the resulting dose-response curve correlates with clinical antibiotic resistant tests – this is important future work, and we have added this in the discussion.

In Figure 1, the y-axis for cell count shown goes only up to 15 but presumably a typical histogram would have orders of magnitude more cells.

- Figure 1 was just a hypothetical example. We have now removed the y axis labels, and revised the figure legend to: "Schematic of an approach to antimicrobial susceptibility testing based on bacterial single-cell phenotypes."

What is a typical proportion of cells not identified by the Mask R-CNN segmenter? It is clear from Figure S11 for instance that many cells are not found.

- The mean Average Recall at an IoU of 0.5 (mAR[0.5]) of the Mask R-CNN was ~60%. The mean average precision and mean average recall plots are now appended Fig S6.

Reviewer 2

The performance evaluation in the manuscript needs to be further specified and improved. For instance, in the context of segmentation, more evaluation metrics, such as mean intersection over union, are required to better demonstrate the performance of Mask R-CNN.

- We agree; mean average precision and mean average recall plots for Mask R-CNN, Cellpose and YOLO have been included as Fig S6).

It would be better to use "precision" and "recall" in the definition of "sensitivity" and "specificity".

- We agree - these have been changed in the manuscript/supplementary information.

The proposed methods in the manuscript lack comparison with other existing methods. Although the authors present the results of Mask R-CNN and DenseNet121, no comparisons with other deep learning models are provided.

- This is an excellent suggestion. In our revision, the classification accuracies of 12 different classification models were explored (including models from the DenseNet, ResNet, and VGG families), alongside all-in-one segmentor/classifiers e.g., Mask R-CNN and YOLO). DenseNet121 produced the highest mean classification accuracy (Fig. R3 below, incorporated in the ms as Fig. S15) and highest end-to-end pipeline classification accuracy across all antibiotic phenotypes (Fig. R1 below, added to the ms as Fig. S16).

In the manuscript, the authors evaluate the robustness of DenseNet121 by training and testing on two class-balanced datasets. However, this approach may not be sufficient to validate the robustness of the proposed method. A better way to evaluate the model's robustness would be to perturbate the input images, such as by adding random noise or cropping, to simulate the variations that occur in real-world applications. The authors should report the changes in performance resulting from such perturbations. If the model can maintain a high per-class accuracy despite these perturbations, it would demonstrate its robustness.

- *The models were already trained with a range of image augmentations randomly applied during training – we have clarified this in the Methods. As you can see from Fig. R4 (added to the revised supplement as Fig. S17), these augmentations can be quite extreme. This prevented the models from over fitting to the data, and resulted in respectable accuracies when evaluating the classification accuracy on unseen data (i.e. the holdout test set). The effect of the augmentation on accuracy can be seen on the training logs, where the accuracy for a given epoch of the un-augmented validation set is ~10% lower than the training set, which is randomly augmented during training (added to the manuscript as Fig. S18).*

The proposed models in the manuscript require further improvement. Firstly, it is worth considering whether it is necessary to use DenseNet121 for classification. Why not use Mask R-CNN to perform segmentation and classification simultaneously, which could potentially improve efficiency? Secondly, the models of Mask R-CNN and DenseNet121 are already widely used in various fields. Therefore, the authors should demonstrate why they choose those two models over other deep learning methods, either by providing a comparison with alternative approaches or by highlighting the unique value of their proposed method in this manuscript.

- As per our response to a similar comment made by Reviewer 1, we have explored several state-of-the-art segmentation and classification models, and different permutations of each, to ascertain the most accurate and generalizable pipeline. DenseNet121 was found to produce highest mean classification accuracy (Fig. R3 below, added to the ms as Fig. S15) and highest end-to-end

pipeline accuracy across all antibiotic phenotypes (Fig. R1 below, added to the manuscript as Fig. S16).

The manuscript contains several careless mistakes. For instance, in the Dose-Response Model fitting section, the authors neglected to italicize α in the phrase "upper $\alpha/2$ critical value." Additionally, citations 38 and 39 are from the same proceedings but have different names.

- Thank you for spotting these mistakes, which have now been corrected in the revised ms.

To aid readers in understanding the scale of different microscopic images of bacteria, it would be helpful for the authors to include scale bars for the following figures: Fig. 2, 3, S1, S3, S4, S8, S9, S11, S12, and S13.

- All figures featuring microscope data have now been updated with scale bars.

To ensure reproducibility, the images used for deep learning training and testing should be made publicly available. I strongly recommend that the authors upload their datasets to an appropriate public data repository. This will allow other researchers to evaluate the performance of the proposed method and compare it with other methods on the same dataset. Furthermore, it will enhance the transparency and credibility of this work, which is essential for the community.

- We agree - the data and models are now publicly available on the Oxford University Research Archive: <https://ora.ox.ac.uk/objects/uuid:12153432-e8b3-4398-a395-abfb980bd84e>; this link has been added to the revised ms.

Statistical analysis should be performed on Fig S7 by adding standard deviations for sensitivity, specificity and accuracy. This can be achieved by K-fold cross-validation or independent repeating experiments. Additionally, it's worth noting that Figures S7 and S10 are tables, not figures, so they should be referred as Tables accordingly.

- We agree the SDs should have been clearly stated in the original manuscript. We have added a table with this data to supplement. Fig. R2 below (added to the revised manuscript as Table S3) shows the variation in classification accuracy across each fold in the K-fold cross-validation experiment, for each antibiotic.
- We have rechecked the manuscript to ensure that Supplementary Tables and Figures are referred to appropriately and consistently.

The authors should consider providing some guidance's for future studies. This would be particularly useful for those who want to employ this model for analytical study or high throughput drug screening.

- Our work represents a proof-of-concept study on four antibiotics and few clinical isolates of *E. coli*, so it would be a bit premature to provide broad guidance for larger and throughput studies. We note that there is significant natural variation in the cellular phenotypes within different experiments and between experiments, so to make the models more robust to these, more data should be acquired from multiple experiments for training/testing. The entire dataset, the python code and the model weights are all provided for readers to explore and apply for their own needs.

Reviewer 3

*I feel the title overstates the findings of the paper. The data generated is only for *E.coli* and a significant proportion of the data generated relates to lab reference strain MG1655. I would suggest this needs to be acknowledged in the title (so include in *E.coli*) as a minimum, but this perhaps needs to go further to say that what is being measured is response(s) to antibiotics, not informing ASTs at this stage and with no evidence that it would work with other bacteria.*

- We overall agree with the suggestions. We thus:
 - reworded the title to reflect the proof-of-concept nature of this work, with the new title being "Deep Learning and Single-Cell Phenotyping for Rapid Antimicrobial Susceptibility Detection in *Escherichia coli*"

- clarified that we measure responses in (sub)cellular morphology to classify cells as susceptible/resistant to an antibiotic.
- highlighted that future work and validation would be required to more robustly evaluate any correlation of our assay to the S/R categories assigned by conventional microbiological AST assays.
- However, we also note that, using ciprofloxacin to treat three strains with a range of MICs, we showed that changes in the ratio of susceptible:resistant cells (measured using our classification approach) correlate very well with bacterial growth inhibition derived using typical clinical testing methods (Fig.6), pointing to the potential of our approach.

The authors refer to other methods being low throughput in the abstract, but make no reference to the likely throughput of the imaging methods described. The time savings are also misleading as they only relate to the time taken to generate antibiotic susceptibility data, not acknowledging that there is a prior culture phase before this.

- We have removed the reference to “low-throughput” in the Abstract, and acknowledge that there are growth-based rapid susceptibility phenotyping platforms (e.g. Accelerate Pheno and Biomerieux Reveal) that perform susceptibility testing much faster than standard growth-based assays. We also modified the text to clarify that time savings would relate only to the susceptibility testing aspect of any diagnostic workflow.

The paper describes responses of lab reference strains MG1655 to 4 antibiotics, with different modes of action. The authors state that rifampicin is clinically relevant but this is not true for E.coli, noting that in the supplementary section they acknowledge that no breakpoint has been determined for rifampicin in E.coli. I would suggest this needs to be made more clear and ideally another antibiotic selected.

- This is a good point. We included rifampicin as it reflected a distinct mode of action (i.e. blocking RNA synthesis). We have thus clarified the rationale for Rif inclusion and stated that only ciprofloxacin, gentamicin and co-amoxiclav are relevant to the management of clinical *E. coli* infections.

The authors also don't make it clear in the text of the results section that the concentrations of antibiotic used for the training experiments are 20-fold higher than the concentration which is used to define clinical resistance.

- We have now explicitly stated the concentration and incubation times for each antibiotic used (Results section); a comparison with EUCAST clinical breakpoints is also presented in Table S1.
- We have also clarified throughout the distinction between:
 - (i) our approach using high concentrations of antibiotics over short incubation times to characterise distinct cellular morphological changes or phenotypes that look “susceptible” versus “resistant” to the effects of antibiotic treatment, and which we used to train our deep learning models, and
 - (ii) S/R categories assigned by detecting growth at a pre-specified susceptibility breakpoint (in this case the EUCAST breakpoint).
- In the final part of our study, however, we clearly demonstrate for three clinical *E. coli* strains with different ciprofloxacin MICs (0.008 mg/L, 0.5 mg/L, 72 mg/L) that the ratio of imaged cells displaying “resistant” and “susceptible” cellular phenotypes after 30 min of incubation with ciprofloxacin at a range of concentrations (0.001, 0.01, 0.1, 0.5, 1, 2, 4, 8 and 16 mg/L) correlates well with bacterial growth inhibition, and could thus represent an effective and “quick-response” proxy for results obtained using standard clinical microbiological growth-based assays.

The authors also don't state (as far as I can see) the MICs that were determined for MG1655 which each of these antibiotics and, as such the concentration used may be very high compared to the MIC and to the MICs which you might expect to see with clinical E.coli strains. If the paper is about ASTs then this appears to be completely counterintuitive to me. The authors need to make it very clear why they use such high concentrations and not the resistance breakpoint MIC, as others have done.

- We have now included a description of the MICs determined for MG1655, which we agree is relevant. We have clarified that the MG1655 *E. coli* strain was incubated with supra-MIC concentrations of ciprofloxacin, gentamicin, rifampicin and co-amoxiclav over 30-60 min timeframes in order to rapidly induce discernible changes in cellular morphology that could be used to train models to recognise cells affected by antibiotic treatment versus those unaffected by treatment (i.e. displaying phenotypes similar to untreated cells) (Fig. S2 and S3).
- For ciprofloxacin, these trained models were then used to detect the proportion of cells classified as having susceptible cellular phenotypes for six clinical isolates with MICs of 0.008 mg/L, 0.03 mg/L, 0.5 mg/L, 8 mg/L, 72 mg/L, and 108mg/L, and following treatment with the same concentration/time combination used to train the model on MG1655 (i.e. 10 mg/L ciprofloxacin for 30 min). This experiment showed that the proportion of cells classified as showing a susceptible cellular phenotype reflected what would be expected based on the MIC – i.e. for the three isolates (EC1, EC2, EC3) with MICs clearly lower than the treatment concentration (i.e. <10 mg/L), >80% of cells were classified as having susceptible cellular phenotypes upon treatment when compared with baseline, whereas for the two isolates (EC5, EC6) with MICs clearly greater than the treatment concentration, there was no discernible difference in cellular phenotypes post-treatment when compared with baseline (Fig.5A). For the isolate with an MIC approximating the treatment concentration (EC4), the classification of cellular phenotypes appropriately reflected this uncertainty (Fig.5A).
- Finally, again for ciprofloxacin, we evaluated the ratios of cells classified as showing susceptible:resistant cellular phenotypes for isolates EC1, EC3 and EC5 (MICs: 0.008, 0.5 and 72 respectively) when incubating each strain for 30 min with nine ciprofloxacin concentrations from 0.001 mg/L to 16 mg/L – mimicking to some extent antibiotic dilution series being evaluated in AST panels used in diagnostic phenotyping, but with much shorter incubation times. This study showed that changes in the susceptible:resistant cellular phenotype ratios (Fig.6A) are appropriately observed at concentrations that mirror both growth inhibition (Fig.6B), and the isolate MIC as determined by E-test (Table S1).

Given that the authors only test 4 antibiotics, its perhaps concerning that the antibiotic class used most commonly in the NHS (represented here by Co-amoxiclav) only gives a very subtle phenotype, despite the very high concentrations of antibiotic used relative to breakpoint. If the message of the paper is that this is a new rapid AST platform, then this observation would appear to make it unsustainable and needs to be addressed by the authors. This is before any evaluation of other beta lactam antibiotics and/or clinical isolates for this class of antibiotics. This is a major concern about the validity of the overall approach.

- We agree that significantly more work needs to be carried out experimentally validate the robustness of such a method for different strains of bacterial species, different bacterial species, and different antibiotics, and have included this as part of a “Limitations” section in the Discussion. We have revised the text substantially to avoid the message that our approach represents a fully-fledged new rapid AST platform; rather, we highlight that our approach holds potential as a new rapid method to characterise antibiotic susceptibility but that further validation and development are required.
- We agree that the co-amoxiclav phenotype is more subtle and that having an effective method for this widely used antibiotic is highly relevant. Important future work as the reviewer highlights will involve evaluating other commonly used antibiotics such as cephalosporins, effects over slightly longer treatment times, and considering cell death as part of cellular phenotyping metrics, but exploring this is beyond the scope of this work.

The authors talk about whole population analysis and this is clearly one of the strengths of the approach. They don't comment on the proportion of the cell population which would need to show a resistant phenotype before the isolate could be classed as resistant, although this is clearly evident from the proportions of the populations that respond to different concentrations of ciprofloxacin (only) in the MIC experiments. This also assumes that all of the strains tested are homo-resistant to ciprofloxacin, which might not be the case. Again this needs to be addressed strongly in the paper, but could be an important feature that advocates for the use of this technology.

- We agree this is a useful addition. We have thus computed the number of observations required to detect antibiotic response phenotypes for each antibiotic for the clinical isolates featured in Fig. 5. For a classification accuracy of 90% with a statistical significance of 1%, the number of observations required from bacteria treated at antibiotic concentrations above the MIC was up to 100-400 unique observations, depending on the strain. This was added to the ms as Table S6.

The statement that the inflection point of the analysis for the MIC determined by the optical imaging is close the clinical AST determined value is perhaps misleading, given that one is a numeric value and the MIC is based on a log2 dilutions series. There is no detail of the AST method used, otherwise it might be appropriate to also determine the inflection point from plotted optical densities in broth dilution assays rather than as for standard assays.

- As the reviewer suggests, we compared three susceptibility metrics for each of the three clinical strains tested, evaluating the impact of a different ciprofloxacin dilutions: (i) using E-testing (i.e. a validated method used in clinical microbiology laboratories); (ii) on growth as measured by looking at the impact of ciprofloxacin concentration on optical densities (see *Methods* for details; Fig.6B for results) and (iii) on susceptible:resistant cellular phenotype ratios (i.e. our imaging/deep learning method).
- As the concentration at which growth (as determined by optical density) was first inhibited (Fig.6B) for each of the three clinical *E. coli* strains (with varying MICs as determined by E-test), correlated with a sharp increase in the ratio of susceptible:resistant classifications as determined by our deep-learning models (Fig.6A), we believe that our method holds promise in inferring isolate MIC. We agree that more work is needed to validate this, and stated this in the Limitations section of the Discussion.

Overall my view is that this is an exciting technology but there are significant gaps in the data provided that do not support its claims to be a potential platform for rapid ASTs. I would suggest considerable revisions of the manuscript to either address these points or to make it clear that this is currently an experimental method only.

- We thank the reviewer for the positive view of our technology and their valuable comments, and we fully agree that our method is not a fully-fledged platform for rapid AST; rather, it is a proof-of-concept method with clear potential for development into a rapid AST platform. A lot more work needs to be carried out to establish this method clinically, and we have substantially revised the text and title to reflect this.

Fig R1. Box plots of the pipeline classification accuracy while utilising different combinations of segmentation and classification models, evaluated over the four antibiotic phenotypes on the holdout test set. The learning rate and batch size were found for each neural network using a grid search. A pipeline featuring Mask R-CNN + Densenet121 was found to have the highest mean accuracy across all antibiotic phenotypes.

Fig R2. Box plots show the variation in K-fold cross-validation classification accuracy for each antibiotic, evaluated on the holdout test set.

Fig R3. Box plots of classification accuracy for a range of different convolutional neural network architectures, evaluated over the four antibiotic phenotypes on the cross-validation test set. The learning rate and batch size were found for each neural network using a grid search. Densenet121 was found to have both the highest mean accuracy, and the most consistent performance across all antibiotic phenotypes.

Fig R4. Example image augmentations that are applied during training. The centre image (highlighted in red) is unperturbed, whereas every other image has been randomly augmented using a random sequence of transformations including horizontal and vertical flips and translations, rotations, cutout as well as Gaussian blurring. The scalebar is 2 μ m.

Fig R5. Example images from the titration dataset shown in figure 6, for the clinical isolate EC1. The measured MIC for the clinical isolate EC1 is 0.008 mg/L.

REVIEWERS' COMMENTS:

Reviewer #2 (Remarks to the Author):

The authors have addressed my concerns and comments satisfactory.